# Incorporating Topological Representation in 3D City Models

Stelios Vitalis *, Ken Arroyo Ohori and Jantien Stoter

3D Geoinformation Group, Delft University of Technology, 2628BL Delft, The Netherlands
* Correspondence: s.vitalis@tudelft.nl

**Abstract:** 3D city models are being extensively used in applications such as evacuation scenarios and energy consumption estimation. The main standard for 3D city models is the CityGML data model which can be encoded through the CityJSON data format. CityGML and CityJSON use polygonal modelling in order to represent geometries. True topological data structures have proven to be more computationally efficient for geometric analysis compared to polygonal modelling. In a previous study, we have introduced a method to topologically reconstruct CityGML models while maintaining the semantic information of the dataset, based solely on the combinatorial map (C-Map) data structure. As a result of the limitations of C-Map's semantic representation mechanism, the resulting datasets could suffer either from semantic information loss or the redundant repetition of them. In this article, we propose a solution for a more efficient representation of geometry, topology and semantics by incorporating the C-Map data structure into the CityGML data model and implementing a CityJSON extension to encode the C-Map data. In addition, we provide an algorithm for the topological reconstruction of CityJSON datasets to append them according to this extension. Finally, we apply our methodology to three open datasets in order to validate our approach when applied to real-world data. Our results show that the proposed CityJSON extension can represent all geometric information of a city model in a lossless way, providing additional topological information for the objects of the model.

**Keywords:** 3D city model; topology; combinatorial map; linear cell complex; CityJSON; CityGML

## 1. Introduction

3D city models have been increasingly adopted in modern analysis of urban spaces, such as the simulation of evacuation scenarios [1] and optimisation of energy consumption for city districts [2,3]. Their key benefit is that they can describe complex 3D geometries of city objects, such as buildings, vegetation and roads; and their semantic information, such as their purpose of use and year of construction.

CityGML is the most commonly used data model for the representation of 3D city models [4], which can be encoded in JSON through the CityJSON data format. The data model incorporates the GML representation, which describes the geometric shapes by their boundaries through a method that is referred to as "polygonal modelling". While polygonal modelling is generally considered a robust representation of 2D data, it has been proven inefficient when representing 3D objects. This reflects to the limited number of 3D processing algorithms that can be easily applied to polygonal modelling [5].

Topological data structures have been introduced in GIS as an alternative to polygonal modelling. Their main characteristic is that they explicitly describe the adjacency and incidence relationships between geometric objects. Those relationships can improve the performance of geometric processing. For example, Maria et al. [6] exploited the topological properties of geometries in order to improve the efficiency of ray tracing in architectural models. Furthermore, topological data structures have the

ability to scale to higher dimensions without adding unnecessary complexity [7]. Therefore, typical GIS operations can be described as dimension-agnostic algorithms which can be applied in an arbitrary number of dimensions. For example, Arroyo Ohori et al. [8] proposed a solution for the extrusion of objects of any number of dimension.

For this reason, we investigated the use of ordered topological structures and, more specifically, combinatorial maps (C-Maps) as an alternative to GML for the representation of geometric information in 3D city models. C-Maps combine the powerful algebra of geometric simplicial complexes with the ease of construction of polygonal modelling [7]. Regarding the practical aspect, they are implemented in a software package as part of CGAL (The Computational Geometry Algorithms Library (http: //www.cgal.org)) and they are efficient with respect to memory usage [9]. While C-Maps originally store only topological relationships between objects, they can be easily enhanced with the association of coordinates to vertices, which results in a linear cell complex (LCC) that incorporate both geometric and topological information.

LCCs based on C-Maps have been used before in 3D city models. Diakité et al. [10] proposed a methodology on the topological reconstruction of existing buildings that are represented through polygonal modelling. They then used the topological information to simplify the building's geometry. This approach is based on the extraction of a soup of triangles from the original geometry which are later stitched together according to their common edges in order to identify the topological relationships. During this intermediate step, the semantic information of the original model, such as hierarchical relationships, are lost as the soup does not retain the information of the original model. Diakité et al. [11] further refined this process and applied it to BIM and GIS models, in order to exploit the topological information to identify specific features of buildings. Although this application includes the reconstruction of CityGML models, it results in a semantic-free model where the original city objects' subdivision is lost.

Previously, we introduced a methodology for the topological reconstruction of CityGML models to LCCs based on C-Maps with preservation of semantics [12]. Because this methodology relies solely on the C-Maps data structure for the representation of all information of the 3D city model, the resulting model suffers from either occasional loss of the semantic subdivision of city object, or a redundancy of information. For example, in that article, we topologically reconstructed the 3D city model of *Agniesebuurt*, a neighbourhood of Rotterdam, which was missing intermediate walls between adjacent buildings. As a consequence of the missing walls, multiple individual buildings were merged under the same volumes in the resulting C-Map. This causes the loss of semantic information of some buildings during the reconstruction as only one city object's information could be attached in the resulting volume.

In this paper, we propose an improved methodology for the topological representation of CityGML models, in order to avoid the limitations of semantics representation in the C-Maps data structure and the limitations of topological representations in CityGML. To achieve that, we integrate the original CityGML data model with C-Maps in order to combine the semantic-representation capabilities of the first with the benefits of a topological data structure. This topologically-enhanced data model is implemented in CityJSON through an extension. We also develop an algorithm in order to transform existing CityJSON datasets. We applied our algorithm to several open datasets to assess the robustness of our method and evaluate the ability of the proposed data model to represent the peculiarities of various datasets.

## 2. Related Work and Background Information

### 2.1. The CityGML Data Model

CityGML is a data model and an XML-based format that has been standardised through the Open Geospatial Consortium (OGC) in order to store and exchange 3D city models [4]. It defines an

object-oriented approach for the representation of city objects, utilising techniques such as polymorphism in order to provide enough flexibility.

In CityGML, a city model contains a number of city objects of different types, all of which inherit from the basic abstract class *CityObject*. Different types of objects can be represented through derived classes: (a) composite objects, such as *CityObject Group*; (b) specialised abstract classes, such as *AbstractBuilding*; or (c) actual city objects, such as CityFurniture and LandUse (Figure 1). Given that a CityGML dataset has a tree structure, the objects can be either listed as immediate child nodes in the model or represented in a deeper layout by grouping objects using the *CityObjectGroup* class.

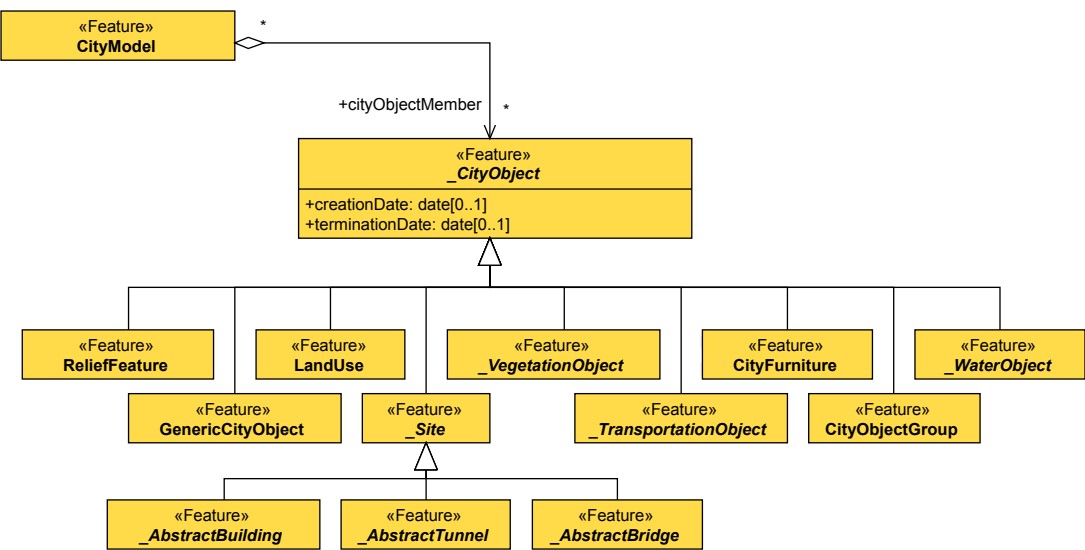

**Figure 1.** The UML diagram that describes CityGML's top level class hierarchy of city objects.

CityGML is a schema that specialises the geographic markup language (GML) [4], therefore it follows GML's geometric representation, which is based on ISO19107, the international standard that defines the spatial characteristics of geometric features [13]. Every *CityObject* contains one or more *Geometry* objects described as GML objects. The *Geometry* object can be extended through composition, therefore a geometry can be a primitive or a composite object of multiple geometries.

According to the CityGML data model, city object semantics are represented through two mechanisms: object types and attributes. The type of a *CityObject* is derived from the class used in order to represent it. Then, additional information for every *CityObject* can be stored in the model through attributes which are described by the CityGML specification. A user can enhance the data model with additional attributes related to their domain requirements, either by using the *GenericAttribute* class or by developing an application domain extension (ADE).

In addition to city object information, CityGML provides a mechanism to further define semantics of individual surfaces that compose the geometry of an object. This is achieved through the *SemanticSurface*, which can semantically describe polygons that bound a geometry. For instance, an individual surface of a building can be defined as a *RoofSurface* to denote that its part of the building's roof. Furthermore, semantic surfaces can be enhanced with attributes in order to attach information to them, e.g., a *WallSurface* can have attributes describing its colour.

## 2.2. The CityJSON Data Format

CityJSON is a data format which uses the JavaScript Object Notation (JSON) encoding in order to implement a subset of the CityGML data model. Its goal is to serve as an alternative to the CityGML data format, which is verbose and complex due to its GML (and, thus, XML) nature. For example, there are at least 26 different ways to encode a simple four-point square in GML (https://erouault.blogspot.com/2014/04/gml-madness.html).

CityJSON uses a simpler structure that allows for less ambiguity and verboseness. First, it uses the JSON encoding, which is easier to parse and write. This is due to its representation, which can be mapped directly to the data structures that are supported by most modern programming languages: key-value pairs (known as maps or dictionaries) and arrays. Second, CityJSON promotes a "flat" list of city objects, while hierarchy can be implied through internal attributes. For example, the `parents` and `children` attributes can be used to associate references between city objects as follows:

```
1  {
2  "type": "CityJSON",
3  "version": "0.9",
4  "CityObjects": {
5  "building01": {
6  "type": "Building",
7  "children": ["buildingpart01"],
8  ... # other properties
9  },
10 "buildingpart01": {
11 "type": "BuildingPart",
12 "parents": ["building01"],
13 ... # other properties
14 }
15 },
16 "vertices": [
17 ... # define vertices
18 ],
19 "appearance": {},
20 }
```

Similar to ADEs for the CityGML data format, CityJSON also provides an extension mechanism for defining domain-specific city objects and attributes. Through CityJSON Extensions, one can introduce new type of city objects or append existing ones with attributes related to the subject of the extension.

### 2.3. Topology in 3D City Models

As mentioned in Section 2.1, CityGML specialises GML, which uses the geometric representation of ISO19107. While GML provides a topology package, CityGML does not deploy it as GML topology is considered "very complex and elaborate" [4]. Instead, it utilises the XML mechanism of *XLinks* in order to re-use common geometric primitives between objects, which can be seen as an implicit representation of adjacency information in the model [14].

Li et al. [15] introduced a topological representation of 3D city models through a proposed CityGML ADE. Their solution offers a way to store topological relationships (such as *touch*, *overlap* and *disjoint*) as decorators of CityGML geometries.

In both approaches, topology is considered as an additional layer of information on top of the geometric representation. Therefore, they do not intend to use topological data structure to represent the objects' shapes.

### 2.4. Linear Cell Complexes and Combinatorial Maps

A linear cell complex is a separable Euclidean space $\mathbb{E}^n$ of non-intersecting cells, where an $i$-cell is homeomorphic to an $i$-dimensional ball and every $i$-cell is linear (i.e., collinear, coplanar, etc.). For example, a three-dimensional linear cell complex denotes zero-cells which are vertices, one-cells which are edges, two-cells which are facets and three-cells which are volumes. A linear cell complex is the special case of a CW complex, whose attaching maps are homeomorphisms and whose cells are polyhedra [16]. Cell complexes are used in applications such as computer graphics, Computer Aided Design (CAD) and geographic in order to discretise 3D geometric objects [17].

A combinatorial map (C-Map) is a data structure that represents an orientable topological partition of $n$-dimensional space [18], and can be used to describe the combinatorial part of a linear cell complex. C-Maps are similar to generalised maps (G-Maps) [19], although the first is capable of representing only orientable cells, while the latter can represent non-orientable cells as well.

C-Maps represent cells in space through *dart* elements. Darts are similar to half-edges (from the half-edge data structure) for an arbitrary amount of dimensions: every part of a directed edge that belongs to an incident $i$-cell ($0 < i \leq n$) is a dart. For instance, when two two-cells (facets) have a common edge, the edge will be described by two darts (one for each facet). By definition, a dart can only belong to one $i$-cell and can be considered as the equivalent of two darts of a G-map associated through $\alpha_0$.

Darts are connected through $\beta_i$ links (where $0 \leq i \leq n$) so that every dart contains one $\beta_i \forall i \in \{0, \ldots, n\}$ (Figure 2). A $\beta_i$ is a link to the next dart in the $i$-cell. For example, in a 4D C-Map, a $\beta_3$ of the dart $d$ links to the dart that belongs to the same edge (one-cell) of the same facet (two-cell) of the same polychoron (4-cell) as $d_1$, but is part of the adjacent volume (three-cell). A *null* dart (denoted as $\varnothing$) is introduced to the C-Map in order to describe the $\beta_i$ of a dart that is not linked to another dart. A dart with $\beta_i = \varnothing$ is called $i$-free.

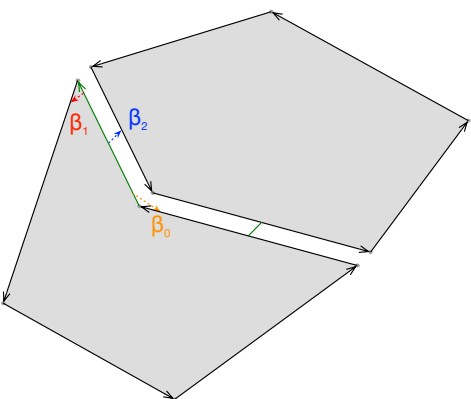

**Figure 2.** Example of a two-dimensional C-Map describing two polygons [20]. The graph is composed of seven zero-cells (vertices), eight one-cells (edges) and two two-cells (facets). These cells are described by darts (denoted as arrows) and their $\beta_i$s (denoted as dashed arrows). Every edge of a facet is described by a dart and $\beta_i$'s are used to describe the links between them: $\beta_1$ is the dart of the "next" edge of the same facet and $\beta_2$ is the dart of the "next" facet of the same edge. $\beta_0$ is a special link to the dart of the "previous" edge of the same face. In this example, only one dart's $\beta_i$s are shown.

To modify C-Maps, we define the *sewing* operation, according to which pairs of corresponding darts of two $i$-cells are linked in one dimension. A $i$-sew operation associates together two $i$-cells along their common incident $(i-1)$-cell. This means that the $\beta_i$'s of every pair of darts along the two $(i-1)$-cells has to be linked.

Cells in a C-Map can be associated to information through a mechanism of attributes. A dart holds a set of attributes, one for every dimension of the C-Map which is called the $i$-attribute of the dart. For example, to set a property of a facet (e.g., colour), one can set this colour value to the two-attribute of every dart of this facet (two-cell).

As mentioned above, C-Maps can describe the combinatorial part of a linear cell complex (LCC). A LCC can be fully described by associating the vertices of the C-Map with points of a $n$-dimensional geometric space (as coordinates) and assuming that all cells of the C-Map are linear. Then, this LCC contains both geometric and topological information for the space.

The C-Map and LCC data structures have been implemented in CGAL as software packages (Figure 3) [9]. Therefore, it is possible to implement software in the C++ programming language which incorporates them.

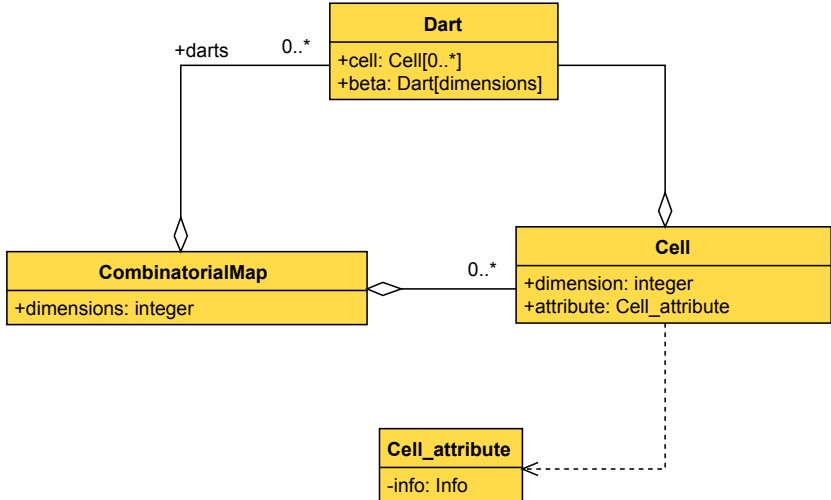

**Figure 3.** The UML diagram that describes the linear cell complex implementation of CGAL.

## 3. Topological Representation of 3D City Models

To represent the topological information of city objects in a CityJSON file, we followed two steps: first, we introduced the LCC entities into the CityGML data model; and, second, we developed a CityJSON extension that provides the necessary encoding instruction in order to store those information in a CityJSON file. We developed the extension definition according to the respective CityJSON specification (https://www.cityjson.org/specs/1.0.0/#extensions). The CityJSON extension definition is available as open source in GitHub (https://github.com/tudelft3d/cityjson-lcc-extension).

### 3.1. Data Model

First, it must be possible to store darts in the city model. Second, every dart has to be linked with: (a) the point that defines the location of the dart's zero-cell; (b) the city object that is associated to the dart's two-cell; (c) the semantic information associated with the dart's two-cell; and (d) its $\beta_i$ darts. It must be noted that $\beta_0$ is not essential for the storage of the map, as it can be implied by the $\beta_1$'s of the structure. Therefore, for a three-dimensional LCC only $\beta_1$, $\beta_2$, and $\beta_3$ are required to be stored.

We decided to associate surfaces (two-cells) with the city objects' semantic information, which might seem a counter-intuitive solution. Initially, volumes (three-cells) might seem as a more suitable match for association with city objects. After all, a city object is normally composed of volumes. Unfortunately, as we proved in [12], that is not always the case. In some city models, there can be multiple city objects that topologically belong to one volume.

While in [12] we proposed a way of forcing three-cells to be divided into multiple volumes, when their two-cells belong to different city objects, the final result is not correct from a topological perspective. The key benefit of using an LCC data structure is to store the topological relationships of a city model's geometry. Consequently, such a solution would largely undermine the benefits of using a topological representation in the first place. Therefore, we decided to associate city objects' semantic information with two-cells in order to be able to maintain the binding of information in cases where one volume is associated with multiple city objects. This way, we ensure a topological representation that is consistent with the geometry and retain a complete association of semantics with the LCC.

We designed a data model that represents the LCC information through the `Dart` class (Figure 4). The class contains the necessary information as attributes: (a) the `vertexPoint` attribute points to the `Vertex` object that stores the coordinates of the zero-cell; (b) the `parentCityObject` attribute points to the `CityObject` associated with the dart's two-cell; (c) the `semanticSurface` attribute links the dart's two-cell with the respective *SemanticSurface* (in case the original polygon has semantic information attached); and (d) the `beta` attribute is a three-element array of `Dart` objects, representing $\beta_1$, $\beta_2$ and $\beta_3$.

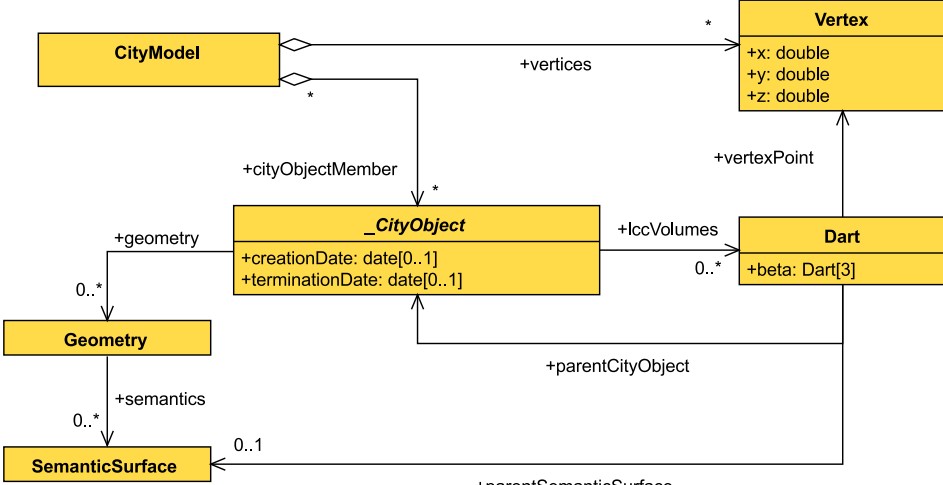

**Figure 4.** The UML diagram that describes the proposed Linear Cell Complex integration with the CityGML data model.

While the `Dart` class and its attributes can represent a complete linear cell complex, they only preserve one-way links from the C-Map to the city model. Nevertheless, it is equally important to be able to identify the three-cells that compose a city object without having to iterate through the whole linear cell complex. This is achieved by introducing the `lccVolumes` attributes in the `CityObject` class, which contains links to the three-cells related to the city object. As it is not possible to directly link to the three-cells, this list contains one of the volume's darts.

### 3.2. CityJSON Extension

We also developed a CityJSON extension in order to implement the data model as described in Section 3.1. This implementation follows the original principles of the CityJSON data format, which aims to be easy-to-use and compact. For this reason, we defined two optimisations in the specification of the extension.

First, we reuse the ''`vertices`'' list as described in the CityJSON specification. This fits perfectly with the requirement of associating points to the zero-cells of the C-Map in order to achieve the linear geometric embedding. Therefore, it is sufficient for the completeness of the final dataset to store the dart information (their betas and parent object associations) and link them to the existing list of ''`vertices`'', instead of introducing a new list.

Second, although in the data model a dart is considered as an object with four attributes (''`betas`'', ''`parentCityObject`'', ''`semanticSurface`'' and ''`vertex`''), such a structure would produce a verbose JSON encoding. That is due to the fact that, for every dart in the dataset, the same attribute names would have to be repeated as keywords. Given the large amount of darts required in order to represent complicated cell complexes such 3D city models, this would result in a burst of the resulting file's size. Instead, we decided to store the four attributes as lists with the same length, equal to the number of darts. This way we avoid the repetition of keywords as they only appear once in the file, thus minimising its size.

### 3.2.1. Darts Representation

To store the darts of the LCC, we added the new ''`+darts`'' root property in the main ''`CityJSON`'' object. It contains four lists containing the values of the respective attributes of the LCCs darts: ''`betas`'', ''`parentCityObjects`'', ''`semanticSurfaces`'' and ''`vertices`''. The ''`+darts`'' object has also the ''`count`'' property which states the number of darts in the LCC. These lists are indexed, so the $n$th element of every list corresponds to the respective attribute of the $n$th dart of the LCC (the lists are one-based numbered, therefore the first element of the list is denoted by the number "1").

According to our implementation, a CityJSON file containing an LCC would contain the following properties:

```
 1  {
 2  "type": "CityJSON",
 3  "version": "0.9",
 4  "CityObjects": {},
 5  "vertices": [],
 6  "+darts": {
 7  "count": 0,
 8  "betas": [],
 9  "parentCityObjects": [],
10  "semanticSurfaces": [],
11  "vertices": []
12  }
13  "appearance": {},
14  }
```

The ''`betas`'' property contains the $\beta_i$'s of the darts. Every element of the list is an array of three integers which refer to the $\beta_1$, $\beta_2$ and $\beta_3$ of the current dart, respectively. In the case this dart is *i*-free, $\beta_i$ is set to $-1$. Otherwise, the $b_i$ refers to the respective dart's index in the list.

The ''`parentCityObjects`'' and ''`vertices`'' properties are single lists. The first is composed of the *ID*s associated with the two-cells of each dart; and the second is composed of the index of the vertex, from the CityJSON's ''`vertices`'' list, associated with each dart's zero-cell.

The ''`semanticSurfaces`'' property associates the two-cell of a dart with a semantic surface of the city model. Every item of this list is an array of two integers, which refer to the indices of the geometry and the semantic surface, respectively, under the parent city object. If the two-cell of a dart does not have a semantic surface associated, then the value of the second value is set to $-1$.

The following is an example of a ''`darts`'' object that would represent an LCC with one triangle:

```
 1  "+darts": {
 2  "betas": [
 3  [2, -1, -1],
 4  [3, -1, -1],
 5  [1, -1, -1]
 6  ],
 7  "parentCityObjects": [
 8  "id-1",
 9  "id-1",
10  "id-1"
11  ],
12  "semanticSurfaces": [
13  [0, 0],
14  [0, 0],
15  [0, 0]
16  ],
17  "vertices": [
18  0,
19  1,
20  2
21  ]
22  }
```

### 3.2.2. CityObject to LCC Association

To be able to efficiently identify the three-cells that compose a city object, we added the ''`+lccVolumes`'' property in the ''`CityObject`''. The ''`+lccVolumes`'' property is a list of darts that belong to the respective three-cells. It has to be noted that not all darts related to a city object are stored in this list; instead, one dart per three-cell is used as an index. Therefore, the number of elements in the list should be equal to the number of volumes associated with the city object.

The following is an example of a city object which is associated to three volumes of the LCC.

```
1  "CityObjects": {
2  "id-1": {
3  "type": "Building",
4  "attributes": {...},
5  "geometry": [...],
6  "+lccVolumes": [0, 5, 28]
7  }
8  }
```

## 4. Topological Reconstruction of 3D City Models

### 4.1. Algorithm

To validate our proposed extension, we developed an algorithm that parses a CityJSON model and appends the LCC information to the model. In [12], we proposed two variations of an algorithm that topologically reconstructs a CityGML model. For the purposes of the research presented in this article, we worked on the basis of the "geometric-oriented" algorithm. We chose this variation because it results in a true topological representation of the model. In addition, our proposed extension does not lose semantic information as it associates two-cells of the resulting LCC with the city model. Therefore, even when multiple city objects are represented under the same three-cell, the semantic association is retained.

We introduced a number of modifications to the original algorithm in order to adjust it towards the requirements of our research. First, the algorithm had to conform with a flat representation of city objects and geometries, as described by the CityJSON specification. Therefore, the recursive call in Algorithm 2 has been removed. Second, we only have to define the essential information that ensures a complete association between city objects and cells, as described in Section 3.2. The resulting methodology is described by Algorithms 1–5.

We have to clarify that the purpose of the reconstruction is to describe the topological relationships between the geometries as described by the original dataset. No modification of the geometry is conducted during the reconstruction in order to ensure the topological validity of the resulting LCC.

Initially, the reconstruction is conducted by the main body (Algorithm 1). This function iterates through the city objects listed in the city model and calls `ReadCityObject` for every city object.

Function `ReadCityObject` (Algorithm 2) iterates through the geometries of the city object. For every geometry, function `ReadGeometry` is called and, then, the created darts' two-attributes are associated to the object's id and geometry's id.

The `ReadGeometry` function (Algorithm 3) parses the geometry by getting all polygons that bound the object. For every polygon, the algorithm iterates through the edges and creates a dart that represents this edge, by calling `GetEdge`. The newly created dart that represents the edge is, then, associated with the semantic surface of the polygon by assigning the respective two-attribute value. Finally, the algorithm accesses the items of index $I_3$ in order to find adjacent three-free two-cells, in which case the two two-cells must be three-sewed. This ensures that volumes sharing a common surface are going to have their adjacency relationship represented in the resulting LCC.

Function `GetEdge` (Algorithm 4) creates a dart that represents an edge, given the two end points of the edge. It gets one dart per point by calling the `GetVertex` function and then conducts a one-sew operation between them. Finally, it iterates through index $I_2$ in order to find adjacent two-free one-cells so that they can be two-sewed.

Finally, function `GetVertex` (Algorithm 5) is responsible for creating darts that represent a vertex in the LCC. The function iterates through the LCC in order to find existing one-free darts with the same coordinates as the point provided; if such a dart is found, it is returned. If no dart is found, the algorithm creates a new dart, associates the coordinates to its zero-attribute, and returns it.

---

**Algorithm 1:** Main body of reconstruction.

**Input:** city model *cm* to be processed
**Output:** linear cell complex *lcc* that contains the geometry and semantics of the provided city
model

1  *R* ← Get all root objects of *cm*;
2  $I_2$ ← An empty index of darts;
3  $I_3$ ← An empty index of darts;
4  *lcc* ← ∅;
5  **foreach** *obj* ∈ *R* **do**
6      ReadCityObject(*lcc, $I_2$, $I_3$, obj*);

7  **return** *lcc*

---

**Algorithm 2:** ReadCityObject

**Input:** linear cell complex *lcc* where the geometry of the city object will be added,
index $I_2$ of 2-free darts in *lcc*,
index $I_3$ of 3-free darts in *lcc*,
city object *obj* to be processed
**Result:** Updates the *lcc* and *I* variables according to the provided city object

1  *G* ← Get all geometries of *obj*;
2  $g_{id}$ ← 0;
3  **foreach** *g* ∈ *G* **do**
4      *D* ← ReadGeometry(*lcc, $I_2$, $I_3$, g*);
5      **foreach** *d* ∈ *D* **do**
6         2-attr-object(*d*) ← id(*obj*);
7         2-attr-geometry(*d*) ← $g_{id}$;
8      $g_{id}$ ← $g_{id}$ + 1;

---

**Algorithm 3:** ReadGeometry.

**Input:** linear cell complex *lcc* where the new two-cell will be created,
index $I_2$ of 2-free darts in *lcc*,
index $I_3$ of 3-free darts in *lcc*,
geometry *g* that corresponds to a set of polygons
**Result:** a new two-cell is created in *lcc* and $I_2$ and $I_3$ are updated accordingly
**Output:** darts *D* that were used for the creation of the two-cell

1  *D* ← ∅;
2  *P* ← Get all polygons of *g*;
3  **foreach** *poly* ∈ *P* **do**
4      **foreach** $v_{cur}$ ∈ *poly* except the last **do**
5         $v_{next}$ ← Get next vertex of *poly*;
6         $d_{new}$ = GetEdge($v_{cur}, v_{next}, I_2$);
7         2-attr-semantic-surface($d_{new}$) ← Semantic surface id of *poly*;
8         push(*D, $d_{new}$*);

9  **foreach** *d* ∈ *D* **do**
10     **if** ∃$d_{op}$ ∈ $I_3$ : *reverse key of d = key of $d_{op}$* **then**
11        Sew(*d, $d_{op}$, 3*);
12        Remove $d_{op}$ from $I_3$;
13     **else**
14        Add *d* to $I_3$;

15 **return** *D*;

---

**Algorithm 4:** `GetEdge`.

---

**Input:** linear cell complex *lcc* where the new edge belongs

　　　　index $I_2$ of 2-free darts in the *lcc*

　　　　vertex $v_1$ that will be the starting point of the edge

　　　　vertex $v_2$ that will be the ending point of the edge

**Output:** dart $d_{new}$ that describes the edge in *lcc*

---

1   $d_{new} \leftarrow$ `GetVertex`$(v_1, 1)$;

2   $d_{next} \leftarrow$ `GetVertex`$(v_2, 0)$;

3   `Sew`$(d_{new}, d_{next}, 1)$;

4   **if** $\exists d_{op} \in I_2 : \textit{0-attr}(d_{op}) = v_2$ **and** $\textit{0-attr}(\beta_1(d_{op})) = v_1)$ **then**

5      `Sew`$(d_{new}, d_{op}, 2)$;

6      Remove $d_{op}$ from $I_2$;

7   **else**

8      Add $d_{new}$ to $I_2$;

9   **return** $d_{new}$;

---

**Algorithm 5:** `GetVertex`

---

**Input:** linear cell complex *lcc* where the output dart belongs

　　　　vertex $v$ to set of the output dart

　　　　degree of freedom $i$ that the output dart must have

**Output:** dart $d$ that belongs to *lcc*, have the vertex $v$ and is $i$-free

---

1   $D \leftarrow$ Get all darts of *lcc*;

2   **foreach** $d \in D$ **do**

3      **if** `0-attr`$(d) = v \wedge \beta_i(d) = \varnothing$ **then**

4         **return** $d$;

5   $d \leftarrow$ Create new dart of *lcc*;

6   `0-attr`$(d) \leftarrow v$;

7   **return** $d$;

---

Algorithms 1–5 call each other in reverse order, meaning that *GetEdge* calls *GetVertex*, *ReadGeometry* calls *GetEdge*, etc. This means that the overall complexity is given by Algorithm 1. Individually, the complexity of the algorithms is as follows:

- The runtime of *GetVertex* depends on the number of darts in the LCC ($d_{lcc}$), i.e. its time complexity is of order $O(d_{lcc})$.

- `GetEdge` calls `GetVertex` twice, which means that the complexity of the first part of the algorithm is $d_{lcc}$ as well. In a reasonable implementation, the sewing operation should be constant time, while the insertion and deletion should be logarithmic, and thus would not change the overall complexity. Because of this, the time complexity is again of order $O(d_{lcc})$.

- `ReadGeometry` iterates through all vertices of the given geometry ($v_g$) and calls `GetEdge`, therefore the first iteration's time complexity is of order $O(v_g) \cdot O(d_{lcc}) = O(v_g\, d_{lcc})$. As the second iteration depends on $d_{lcc}$, which does not change the order of complexity, the final time complexity of `ReadGeometry` is that of the first iteration: $O(v_g\, d_{lcc})$. This assumes the same logarithmic access to the index and constant time iteration from one dart to another (for sewing and setting attributes).

- `ReadCityObject` iterates through all geometries of a city object and calls `ReadGeometry` for every one, which means that the previous calls to `GetEdge` are applied to every vertex of the city object (as opposed to those of a single two-cell). Therefore, the algorithm depends on the size of all vertices of all geometries ($v_{cityobject}$) and in accordance with `ReadGeometry` complexity it is of order $O(v_{cityobject}\, d_{lcc})$).

- Finally, the main iteration repeats `ReadCityObject` for every city object. Following the same logic as before, `GetEdge` ends up being called for every vertex of the city model ($v_{citymodel}$). Therefore, the time complexity of the algorithm is of order $O(v_{citymodel} \ d_{lcc})$.

*4.2. Implementation*

We implemented the proposed algorithms in computer software using the C++ programming language. We used the CGAL LCC package (https://doc.cgal.org/latest/Linear_cell_complex/index.html) for the data structure that keeps the topological information. JSON for Modern C++ (https://github.com/nlohmann/json) by Niels Lohmann was used for CityJSON.

The tool created is a command-line application that is available under the MIT license in a public repository (https://github.com/tudelft3d/cityjson-lcc-reconstructor). It creates an executable file that can be provided with an existing CityJSON file and create an LCC that represents its geometry. The resulting LCC can be saved either as a CGAL C-Map file (`.3map`) or as a new CityJSON.

## 5. Validation of Methodology

To verify the completeness of our proposed methodology, we applied it to three open datasets and visualised their topological and semantic information.

*5.1. Datasets*

We used the software described in Section 4.2 in order to reconstruct three existing open dataset available as CityJSON files:

- Den Haag dataset of buildings and terrain (https://data.overheid.nl/dataset/48265-3d-lod2-stadsmodel-2010-den-haag-citygml),
- Rotterdam's Delfshaven dataset of buildings (http://rotterdamopendata.nl/dataset/rotterdam-3d-bestanden), and
- A dataset representing a landscape around a railway, originally introduced to demonstrate a plethora of CityGML 2.0 city object types.

5.1.1. Den Haag

This is a dataset of buildings and the terrain provided by the municipality of the Hague. The model was created in 2010 and is based on the aerial photos acquired and the registration of buildings (BAG (https://zakelijk.kadaster.nl/bag)) of that year. The dataset concerns around 112,500 buildings of the municipality of the Hague and the neighbouring municipalities, divided into 152 tiles.

We tested our methodology against the first tile of the dataset, which is available as example dataset for CityJSON (http://www.cityjson.org/en/0.9/datasets/). The file contains 2498 city objects, of which one is a `TINRelief` and the rest are `Building` objects. It contains 1991 LOD2 geometries of `MultiSurface` and `CompositeSurface` type, with semantic surfaces `RoofSurface`, `WallSurface` and `GroundSurface` present.

5.1.2. Delfshaven

This is the first version of Rotterdam's 3D city model, which was released as open data in 2010. It was created based on the basic topolographical map of the Dutch Kadaster (BGT (https://zakelijk.kadaster.nl/bgt)) and LiDAR data for the extrusion of the buildings. It is divided into 92 files separated according to the municipalities neighbourhood administration boundaries.

In the research described in this article, we worked with the CityJSON file containing the Delfshaven neighbourhood. The file contains 853 building objects of LOD2 and a respective number of `MultiSurface` geometries with three semantic surfaces present: `RoofSurface`, `WallSurface` and `GroundSurface`.

In this dataset, the walls between adjacent buildings are missing. This causes multiple buildings to merge under one volume, topologically, instead of being individual volumes one next to the other.

### 5.1.3. Railway Demo

This datasets is a procedurally produced 3D city model with the intention to demonstrate most of CityGML 2.0 city object types. It is available in CityJSON format and contains 121 city objects of fourteen different types. It also contains 105 `MultiSurface` and `CompositeSurface` geometries with semantic surfaces.

### 5.2. LCC Viewer

We built a viewer to evaluate the topologically reconstructed CityJSON files (https://github.com/liberostelios/lcc-viewer). Our viewer's source code is based on CGAL's demo 3D viewer of the LCC data structure. We added two features that we needed for our experiments: (a) the ability to load CityJSON files with an LCC; and (b) an option to render surfaces, thus objects, in three different ways: per volume, per semantic surface type and per city object id.

### 5.3. Reconstruction and Evaluation of Datasets

Using our software (Section 4.2), we topologically reconstructed the three datasets and created three CityJSON files containing the LCC according to the proposed extension (Section 3.2). The characteristics of the resulting datasets is shown in Table 1.

**Table 1.** Statistics of the datasets that were reconstructed from our software.

| Dataset | File Size (MB) | | Number of | | | | | | |
|---------|----------------|-------|--------------|------------|---------|------------|---------|---------|---------|
| | Original | Final | City Objects | Geometries | Darts | Zero-Cells | 1-Cells | 2-Cells | 3-Cells |
| **Den Haag** | 3.00 | 8.56 | 2498 | 1991 | 84,795 | 24,835 | 42,658 | 21,804 | 1991 |
| **Delfshaven** | 2.60 | 7.08 | 853 | 853 | 68,500 | 26,782 | 42,644 | 15,484 | 1192 |
| **Railway demo** | 4.31 | 18.92 | 121 | 105 | 243,491 | 76,821 | 135,514 | 65,196 | 5789 |

The resulting datasets have significantly grown in size after the reconstruction. We identified that the main factor of growth is the number of darts, which seemed to contribute consistently on the added space of the resulting file. On average, the three datasets required about 65 bytes per dart.

To verify the complete representation of semantics and cells association, we inspected the final datasets in our viewer (Section 5.2). Every dataset was visualised with three different facet colouring methods: per individual volume, per semantic surface type and per city object id (Figure 5). The per-volume facet formatting highlights the topological characteristics of the dataset. Using the per-city-object facet formatting, we verified that the association between the semantics of the dataset and the cells of the LCC are retained and complete. Finally, using the per-semantic-surface formatting, we verified that association between surfaces and its semantic information in the respective geometry of the original model were also maintained.

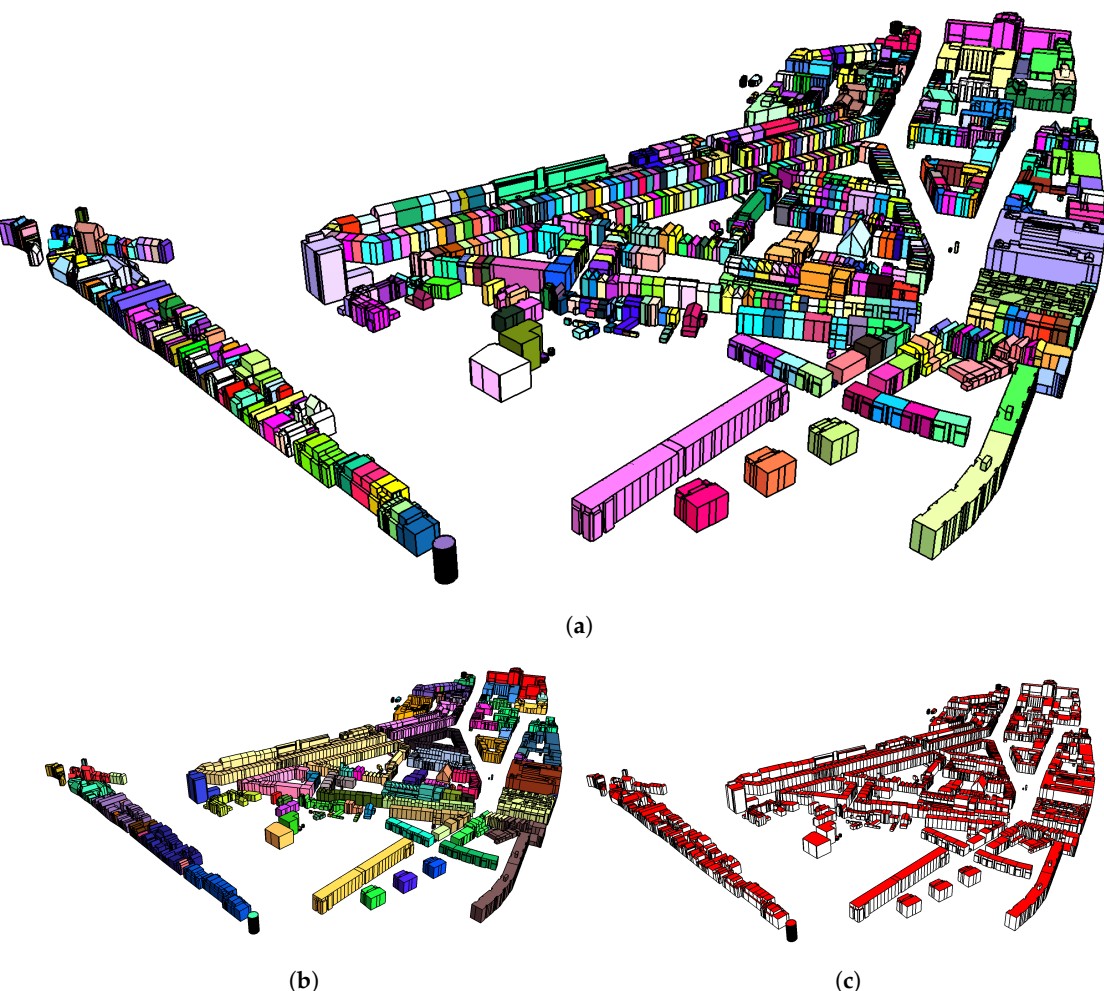

(a)

(b)                                                                    (c)

**Figure 5.** The Delfshaven dataset visualised in the LCC viewer according to three different colour formatting options. (**a**) Every facets (two-cells) is coloured according to the city object in the CityJSON structure. This figure proves that buildings that are merged in the same volume (three-cell) maintain their association with the original city objects. (**b**) All facets (two-cells) that are incident to the same volume (three-cell) have the same colour. This figure highlights the topological characteristics of the dataset and shows that multiple semantically individual buildings have been merged, topologically, under the same volume. (**c**) Every facet (two-cell) is being coloured according to the semantic surface related to it: *red* highlights roof surfaces and *white* highlight walls. This figure proves that the the resulting dataset maintains the association between facets (two-cells) and semantic surfaces in the CityJSON structure.

The inspection proved that the proposed CityJSON extension is complete enough to provide the association between semantics and cells. The viewer successfully highlighted the datasets according to both topology (volumes) and semantics (city object ids and semantic surfaces).

During the inspection of the statistics and the visualisation, we identified a degenerate case. We would have expected the Delfshaven dataset to have fewer volumes (three-cells) than city objects, given that multiple buildings where merged during the reconstruction. Nevertheless, that did not occur as, according to Table 1, the number of three-cells was greater than the number of city objects. After further investigation of the model, we identified two reasons for this: first, a small number of the three-cells was composed by single facets which were topologically invalid with their surroundings, therefore they were not merged to the same volume as the rest of the surfaces of those objects; and, second, a great number of three-cells was "noise" in the LCC, as they were single edges without

surface or volume (Figure 6). We verified that those edges were present in the initial model as zero-area surfaces and they were retained in the resulting LCC.

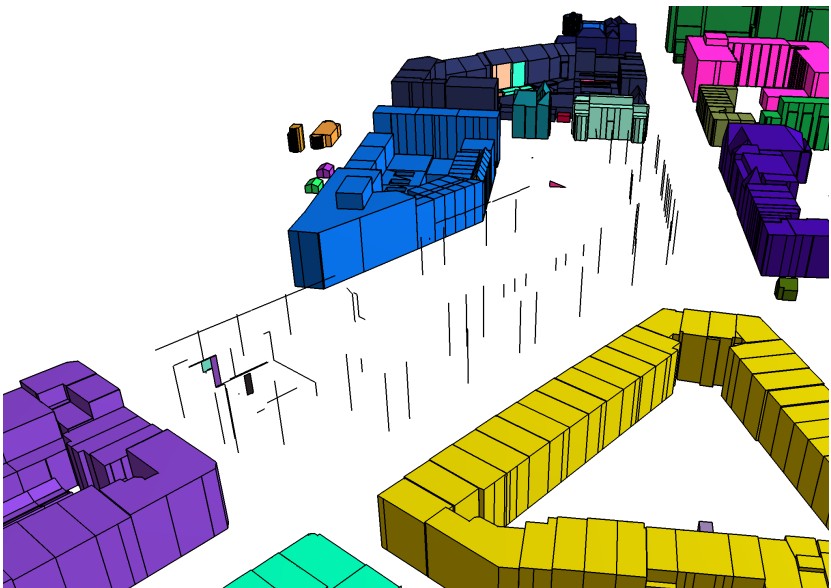

**Figure 6.** Topologically invalid surfaces and "noisy" single-edged volumes were identified in the Delfshaven LCC when the buildings of the area where hidden.

## 6. Conclusions and Discussion

In this article, we propose a topological representation for 3D city models by incorporating an LCC based on the C-Map data structure in the CityGML data model. We materialised this solution by developing an extension for CityJSON and the respective algorithms that can compute the topological links based on the geometry of an existing dataset. Furthermore, we implemented this solution in computer software and applied it to three open CityJSON datasets to evaluate the completeness of the proposed solution.

We incorporated the C-Map data structure as its implicit representation of cells, through darts, is optimal with regard to storage space. As a consequence of its implicit representation, a C-Map cannot be associated with city objects according to the suggestions of ISO19109. This is because ISO19109 assumes that the topology of a feature (i.e., city object in this context) is described through an explicit representation, such as the one described in ISO19107.

We need to clarify that Algorithm 3 does not factor holes (i.e., internal rings) in during the reconstruction. This is because C-Maps cannot represent holes of two-cells. A potential solution would require altering the geometries (e.g., by inserting an edge between outer and inner rings of a polygon). Nevertheless, such an operation is outside the scope of this research.

In addition, the algorithms provided do not distinguish geometries according to their LoDs. This means that, in the case of a multi-LoD dataset, the resulting LCC will represent the geometries of different LoDs. While in some cases this might not be an issue (e.g., if different objects have different LoDs), in cases where one object has more than one LoDs, the resulting LCC would contain both versions. Such an LCC can be perceived as a wrong representation of the city model's geometry. To deal with these cases, in our implementation, we allow users to filter the geometries by LoD. Nevertheless, this option is not present in Algorithm 2. This is because we believe that every application might have different requirements, therefore we leave it to one's discretion to alter the respective algorithm according to their needs.

Furthermore, as mentioned in Section 4, the algorithm does not modify the original geometries in order to ensure the topological validity of the resulting LCC. That is because the purpose of this methodology is to represent the topological relationships of objects exactly as described by the original

dataset. For example, two adjacent buildings sharing the exact same common wall will be represented as three-linked in the resulting LCC, but that is not the case if they partially share a surface. That would require some kind of a topological clean-up operation, which would ensure the topological validity of the input. Nevertheless, this is outside the scope of this research and can be further researched independently.

Our results show that it is possible to represent any CityGML dataset based on the C-Map data structure without missing semantic information from the original dataset. In addition, the proposed two-way linking mechanism between the entities of the data model and the LCC provides access to the efficient resulting 3D city model based on either a semantic-oriented traversal (by iterating through every city object of the model) or a geometric-oriented traversal (by visiting all darts of the LCC).

We believe that our solution can provide useful information for the geometrical processing of 3D city models. For example, it can assist on the repair of invalid geometries, such as non-watertight solid, based on the existence of two-free darts in the LCC. In addition, our findings regarding "noisy" three-cells in the Delfshaven dataset (Section 5.3) proves that LCC statistics can provide useful insights for the identification of invalid or erroneous data. However, repairing of invalid geometries is outside the context of this research. Therefore, overlapping cells should be expected in the resulting LCC, as soon as the source 3D city model geometries are not topologically valid.

In the future, we intend to utilise this CityJSON extension in order to conduct analysis on the topological matching of existing multi-LoD datasets. We are also planning to use the proposed data structure in order to represent those datasets in four dimensions.

**Author Contributions:** Conceptualization, S.V., K.A.O. and J.S.; Investigation, S.V.; Methodology S.V. and K.A.O.; Validation, S.V. and K.A.O.; Visualization, S.V.; Writing—original draft, S.V.; Writing—review & editing, K.A.O. and J.S.

**Funding:** This project received funding from the European Research Council (ERC) under the European Union's Horizon 2020 research and innovation programme (grant agreement No. 677312 UMnD).

**Acknowledgments:** We would like to thank Tom Commandeur for his assistance in the optimisation of our topological reconstruction software and Kavisha for her insights in the design of UML diagrams describing our data model.

**Conflicts of Interest:** The authors declare no conflicts of interest.

## Abbreviations

The following abbreviations are used in this manuscript:

| | |
|---|---|
| C-Map | combinatorial map |
| LCC | linear cell complex |
| GML | geography markup language |
| ISO | International Organization for Standardization |
| BIM | building information modelling |
| GIS | geographic information system |
| JSON | JavaScript Object Notation |

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
