# Peer review of "Incorporating Topological Representation in 3D City Models"

_ijgi, doi:10.3390/ijgi8080347_

Round 1

Reviewer 1 Report

Topic of the manuscript ‘Incorporating topological representation in 3D City Models’ is the extension of 3D city models (CityGML models) by topological structures. A proposal for extension is presented on a conceptual level (UML diagrams) and on an implementation level focusing on the language CityJSON. The topic of extending 3D city models with topology is of high interest for readers of IJGI.  The manuscript is well written. In my point of view, there are some deficiencies or unclear points which require a revision of the manuscript:

•    One of the main characteristics of CityGML is that a city object (feature) has different geometries in different Levels of Detail with different names. Hence, a feature may have (and has in real data sets) multiple geometries. With regard to topology, only geometries of the same Level-of Detail interact. This important aspect of CityGML is completely ignored in the manuscript. A paper where “CityGML” appears in the title must deal with this aspect.

•    There is an ISO standard 19109 “Rules for application schema” which defines on a conceptual level (UML diagrams) how features should be linked with geometry and topology.  Why did the authors not apply this standard? It is not even mentioned in the manuscript. It has always advantages to apply existing standards if they are available.

•    Figure 3: Linking between CityGML and topology is unclear. In the text, you define semantic surface as Roof surface, Wall surface, etc. In CityGML, such boundary surfaces are CityObjects as well. So, what is the relation between _CityObject and semantic surface in the uml diagram? Which of the multiple geometries do you mean? (see first remark)

•    Page 3, line 93: It is not true that GML 3 is based on the simple feature (SF) specification. GML is a completely different standard defining 3D-objects (solids), topology, complexes, composites and different aggregation concepts. All SF specifications do not provide these concepts. SF as defined by ISO does not even define 3D coordinates.

•    Most readers of IJGI are not familiar with the CityJSON notation and its concepts. Are there references between objects?  In order to give an impression, an example should be given in sec. 3.2.

•    Page 4, line 119: a C-map is introduced as topological partition of space. Is there such a concept as an outer, unbounded 3-cell in your model?

•    Is the notion of LCC introduced in this paper? There is no reference.

•    Algorithm 1 – 5: the algorithm cannot deal with polygons with interior rings / holes, right? If so, this should be mentioned. I did not see where overlapping edges (resulting in splits) or overlapping polygons (e. g., a building with adjacent garage, resulting in splits of polygons) are detected. There are no geometrical operations applied in alg. 1 – 5. If these cases are not covered, the topology is of very limited use. Please clarify. Where are 3-cells generated in alg. 1-5?

•    Typos: page 8, line 290: edges, creates

Author Response

thank your for your valuable comments, we have revised the paper according to your comments and you can see the details by the attachment.

Reviewer 2 Report

In this paper, the authors described how they integrate geometry, topology and semantics of the city model by implementig a CityJSON extension to encode the C-Map data. Paper is well written and shows good examples of use.

Several suggestions are recommanded in order to improve the quality of the paper and be more communicative to the reader. These are summerized as follows:

·        Related work should be enriched

·        Put figure 1 after it is first mentioned

·        line 36 –dot on a wrong place

·        page 3 – link not working http://www.cityjson.org/en/0.9/extensions/

·        you never referenced Figure 2. Word Figure when referencing should be capitalized where it isn’t (later in text)

·        does these algorithms solved all previous problems, or are some of them still remain?How you propose repairing of invalid geometries?

·        Line 391 – something’s wrong, lines instead of space

·        Missing title for References

Author Response

Point 1: Related work should be enriched

We enriched the related work in multiple ways: a) a figure to explain C-Maps was introduced; b) LCCs are now explicitly introduced and better associated with C-Maps, as well as a short comparison with G-maps was added; and c) a subsection regarding topology in 3D city models was added. As a result, the following references were added: 13, 14, 15, 16, 18 and 19.

Point 2: Put figure 1 after it is first mentioned

Fixed.

Point 3: line 36 –dot on a wrong place

Fixed.

Point 4: page 3 – link not working http://www.cityjson.org/en/0.9/extensions/

The link was changed due to CityJSON 1.0 release. We have updated the link to reflect that.

Point 5: you never referenced Figure 2. Word Figure when referencing should be capitalized where it isn’t (later in text)

We have added a reference to Figure 2 and fixed capitalisation of all references (figures, sections and algorithms).

Point 6: does these algorithms solved all previous problems, or are some of them still remain?How you propose repairing of invalid geometries?

We enriched the discussion to better reflect limitations and the scope of the research. Repairing of invalid geometries is outside of the scope of this research and now this is explicitly stated.

Point 7: Line 391 – something’s wrong, lines instead of space

We have used em-dashes but we acknowledge that this might cause confusion to the readers. We now use parenthesis, instead.

Point 8: Missing title for References

Fixed

Reviewer 3 Report

The article "Incorporating topological representation in 3D City Models" shows how to extract a topological model from CityGML data. As CityGML is a very verbose XML format which does is not directly accessible to topological queries, this research idea is very welcome and helpful for the scientific community. However, there are some methodological issues which need to be resolved in order to make this article publishable:

Combinatorial maps. From their definition, it is not clear for me, what exactly a combinatorial map is. Is it related in some way to the concept of G-maps by Pascal Lienhardt (1994)? After clarifying the definition (what is a dart, what is a cell?), the visualisation of a small example would be nice in order for the reader to grasp the idea of 'dart' and 'n-cell' in this context.

Linear Cell Complex. The 'linear' part is understandable, although not clearly defined. But the term 'cell complex' should be clarified. In mathematics, this is actually not used. Topologists say 'cw complex' instead. Is that what is meant? In particular, there is a boundary operator which lacks its mentioning and definition, in order to see how the cells made from darts relate to each other in their boundaries (what is the boundary of a cell?).

Topological consistency in Section 3.1. How is this notion defined? Notice that there are many different definitions in the literature.

Section 3.2.2. Index for storing 3-cells by taking a dart. Observe that there may be one dart related to more than one 3-cell. This would mean that the index is ambiguous. It should be explained how this case is dealt with.

Algorithm. An estimation of the time complexity of the algorithms 1,2,3,4,5 and the overall methodology is missing.

For these reasons, I recommend a major revision.

Details.

l. 66. 'building' -> 'buildings'

l. 94. 'They Geometry' -> 'The Geometry'

l. 139. 'with point of' -> 'with points of'

l. 154. 'LCC beta_1 ... are only required' -> 'LCC only beta_1, ... are required'

l. 170. cf. Remark 4.

l. 227. The plural of 'index' is 'indices'

l. 256. 'a city objects' -> 'a city object'

l. 257. cf. Remark 4.

l. 273. 'on the base of' -> 'on the basis of'

l. 281. 'that ensure' -> 'that ensures'

Algorithm 3. 'geometry g that a set of polygons'. I do not understand what this means.

l. 371. 'less volumes' -> 'fewer volumes'

l. 381. 'a LCC' -> 'an LCC'

Author Response

Point 1: Combinatorial maps. From their definition, it is not clear for me, what exactly a combinatorial map is. Is it related in some way to the concept of G-maps by Pascal Lienhardt (1994)? After clarifying the definition (what is a dart, what is a cell?), the visualisation of a small example would be nice in order for the reader to grasp the idea of 'dart' and 'n-cell' in this context.

We have altered the text of the definition (also with respect to change for point 2). We have added a small reference to G-maps and a figure with an example of a two-dimensional C-Map to explain the concept.

Point 2: Linear Cell Complex. The 'linear' part is understandable, although not clearly defined. But the term 'cell complex' should be clarified. In mathematics, this is actually not used. Topologists say 'cw complex' instead. Is that what is meant? In particular, there is a boundary operator which lacks its mentioning and definition, in order to see how the cells made from darts relate to each other in their boundaries (what is the boundary of a cell?).

We altered the text to better explain what a linear cell complex is and how a C-Map can describe its combinatorial part. A respective reference was added, as well.

Point 3: Topological consistency in Section 3.1. How is this notion defined? Notice that there are many different definitions in the literature.

That was an unfortunate use of the term "topological consistency". We re-phrased the sentence to "...we ensure a topological representation that is consistent with the geometry...", instead.

Point 4: Section 3.2.2. Index for storing 3-cells by taking a dart. Observe that there may be one dart related to more than one 3-cell. This would mean that the index is ambiguous. It should be explained how this case is dealt with.

By definiton, a dart can only be related to one i-cells (i for every dimension).

Point 5: Algorithm. An estimation of the time complexity of the algorithms 1,2,3,4,5 and the overall methodology is missing.

We have added a paragraph where we estimate the time complexity of the algorithms and the complete solution.

Point 6: Details

All comments incorporated except for those related to point 4, which did not require fixing.

Round 2

Reviewer 1 Report

Some of my concerns have been addressed, at least in a satisfactory way. Two issues have still not been addressed: Point 3 and Point 8 (Geometry, holes have been addressed). For Point 3, the text has not been modified. The term “semantic surface” and its relation to “CityObject” is still unclear. What is “the semantic information of the incident surface”? In CityGML, a geometric surface may have different semantic tags, e.g. a polygon may be representing a Building and a roof. Describe the link to CityGML precisely.

Point 8: If I understand it correctly, your algorithm derives topological information only if the corresponding geometrical elements which touch are completely identical. This is in mostly not the case. Consider an ordinary (urban) CityGML dataset with incident buildings and building parts which touch each other. These incidence relations are not detected by your algorithm and hence, not present in your topology. Such a “topology” is of very limited use. It is restricted to the isolated relations between the components of a solid representing a building. For analysis purposes as mentioned in the introduction, such a topology is inadequate. In my point of view, your algorithm does not “reconstruct” topology, as you claim in the introduction. This limitation of the algorithm has to be discussed and stated prominently in the paper (introduction and main part).

Author Response

Point 1: The term “semantic surface” and its relation to “CityObject” is still unclear. What is “the semantic information of the incident surface”? In CityGML, a geometric surface may have different semantic tags, e.g. a polygon may be representing a Building and a roof. Describe the link to CityGML precisely.

In order to better explain the concept of semantic surfaces we have introduced a specific paragraph in the CityGML data model description, under related work. We have also rephrased the respective part related to the association between darts and their respective 2-cell's semantic surface information.

Point 2: If I understand it correctly, your algorithm derives topological information only if the corresponding geometrical elements which touch are completely identical. This is in mostly not the case. Consider an ordinary (urban) CityGML dataset with incident buildings and building parts which touch each other. These incidence relations are not detected by your algorithm and hence, not present in your topology. Such a “topology” is of very limited use. It is restricted to the isolated relations between the components of a solid representing a building. For analysis purposes as mentioned in the introduction, such a topology is inadequate. In my point of view, your algorithm does not “reconstruct” topology, as you claim in the introduction. This limitation of the algorithm has to be discussed and stated prominently in the paper (introduction and main part).

We understand the confusion regarding this part. The algorithm does recognise adjacency relationships between volumes (e.g. touching buildings), but only in the case of an exact common surface between them. In order to further clarify the scope of the topological reconstruction we improved the text in several places. First, we have further clarified how the 3-linking operation works in the description of the ReadGeometry algorithm. Second, we have added a paragraph in the introduction of Section 4.1 which explicitly states that no geometry modification is conducted during the reconstruction. Third, we have added a new paragraph in the conclusions which acknowledges the limitations of the reconstruction and suggests that a topological clean-up operation can be incorporated in the algorithm as part of another research

Reviewer 3 Report

Most issues raised by me are properly addressed, in particular, the definition of linear cell complex is clarified. Please also state that a linear cell complex is the special case of a cw complex, whose attaching maps are homeomorphisms and whose cells are polyhedra. The definition of cw-complex can be found e.g. in

Hatcher, A. Algebraic Topology. Cambridge University Press, Cambridge MA, 2002

Unfortunately, there is still a major problem with combinatorial maps. The definition of a dart remains unclear, and I do not have the book [17]. So, what do you mean by "part of an edge that belongs to every possible combination of i-cells"? Figure 2 is not helpful, either. But Wikipedia (english) has a general definition of combinatorial map, and I see the difference to G-map. Why do you not take this (if it is equivalent to what you do mean)?

In C-maps, darts are somehow related to something called 'cells'. Please provide a definition for 'cell' in a C-map. Otherwise, I cannot check if your idea of using one dart per 3-cell as an index is well-defined. Important for this idea to work is that no two distinct 3-cells share a common dart. So, please clarify.

Next, you associate with a linear cell complex K a C-map X. How? In particular, how are the darts of X constructed from K? What about the definition of beta_i? Are the conditions on beta_i all satisfied in order for X to be a well-defined C-map?

Time complexity. The complexities for the individual algorithms seem correct. Yet, an upper bound for the overall time complexity is not given in terms of cells:

The overall time complexity seems to have an upper bound

O(v_g x v_cityobject x v_citymodel x d_lcc)

which could probably be simplified by relating everything to the total number of vertices.

However, d_lcc should be related to the number of cells in citymodels, in particular in view of
the result of

P.E. Bradley and N. Paul. Comparing G-maps with other topological
data structures. Geoinformatica (2014) 18:595–620.

which says that the number of darts in G-maps grows exponentially with the dimension. Still, C-maps might have fewer darts than G-maps (I do not know), and the dimension is only 3. But still, there might be many darts, if you consider that in G-maps a dart is a cell-tuple

    Volume-Face-Edge-Vertex

and there are many faces, edges and vertices at the boundary of a volume.

So, please give an upper time-complexity bound in terms of cells.

For the reasons above, I recommend another major revision.

Author Response

Point 1: Please also state that a linear cell complex is the special case of a cw complex, whose attaching maps are homeomorphisms and whose cells are polyhedra. The definition of cw-complex can be found e.g. in

Hatcher, A. Algebraic Topology. Cambridge University Press, Cambridge MA, 2002

We have added this sentence and reference to the revised text.

Point 2: The definition of a dart remains unclear, and I do not have the book [17]. So, what do you mean by "part of an edge that belongs to every possible combination of i-cells"? Figure 2 is not helpful, either. But Wikipedia (english) has a general definition of combinatorial map, and I see the difference to G-map. Why do you not take this (if it is equivalent to what you do mean)?

In C-maps, darts are somehow related to something called 'cells'. Please provide a definition for 'cell' in a C-map. Otherwise, I cannot check if your idea of using one dart per 3-cell as an index is well-defined. Important for this idea to work is that no two distinct 3-cells share a common dart. So, please clarify.

Next, you associate with a linear cell complex K a C-map X. How? In particular, how are the darts of X constructed from K? What about the definition of beta_i? Are the conditions on beta_i all satisfied in order for X to be a well-defined C-map?

We have refined the text in order to further improve the definition of C-Maps and its relationship to G-maps and LCCs:

A better explanation of how cells are part of C-Maps and how they are described by darts is now introduced.

The definition of darts has been slightly alterted to reflect their similarities with half-edges and looks closer to the definition of C-Maps in the CGAL documentation.

A simple two-dimensional example of two faces is given in the introduction of darts.

A sentence to specify that a dart can only belong to one cell has been added. It also clarifies the similarity between darts of C-Maps and G-maps.

Figure 2 has a more detailed explanation which should help the reader better understand the concept of darts and beta_i's.

The text that links C-Maps and LCCs has been revised to better explain their connection and how the first can describe the combinatorial part of the latter.

Point 3: Time complexity. The complexities for the individual algorithms seem correct. Yet, an upper bound for the overall time complexity is not given in terms of cells:

The overall time complexity seems to have an upper bound

O(v_g x v_cityobject x v_citymodel x d_lcc)

which could probably be simplified by relating everything to the total number of vertices.

However, d_lcc should be related to the number of cells in citymodels, in particular in view of the result of

P.E. Bradley and N. Paul. Comparing G-maps with other topological data structures. Geoinformatica (2014) 18:595–620.

which says that the number of darts in G-maps grows exponentially with the dimension. Still, C-maps might have fewer darts than G-maps (I do not know), and the dimension is only 3. But still, there might be many darts, if you consider that in G-maps a dart is a cell-tuple

Volume-Face-Edge-Vertex

and there are many faces, edges and vertices at the boundary of a volume.

So, please give an upper time-complexity bound in terms of cells.

We agree to the initial comments of the reviewer regarding the upper bound, which is already incorporated in our final calculation of the overall complexity ( O(v_citymodel d_lcc) ). While the reference does provide a solid ground on the relationship between the number of darts and dimension, in our case the dimension is fixed (three) and the number of darts cannot be directly related to the amount of cells (i.e. faces per volumes and vertices per face). That is because the number of darts is heavily influced by the configuration of cells and not their quantity.

Round 3

Reviewer 3 Report

This second revision of the manuscript meets some of my concerns. However, two important ones are not properly addressed. I fully understand that this is about applications of topology in city modelling, and not about topology itself. However, without a rigourous definition of the objects used in the study, it is impossible to follow the arguments. I apologize for not having made this sufficiently clear in my last review.

To be specific, my first concern is in the definitions of the following terms:

- "linear". This is a property of mappings, not of cells. Therefore, the statement "every i-cell is linear" does not make sense. Please state that you mean that each i-cell is the interior of an i-dimensional polytope, if this is what you mean.

-  "part of a directed edge". I can only interpret this as meaning "subset" or "coordinate". Any other meaning either does not make sense or must be clarified. We know that a directed edge in a graph is a pair (v,w) of vertices. So, you could mean v or w. But why not then just say "vertex incident to an edge"? This is clearer. But I guess, this is not what is meant.

- "belongs to". The rigourous meaning of this expression is also unclear.

- "dart". I have read the explanation of CGAL. It definitively lacks rigour, as it does not answer the question: "What IS a dart?" I hope, that your main reference [17] has a rigourous definition. Anyway, the Wikipedia article on "Combinatorial Maps" has one. I hope that the definition there is equivalent to the definition in [17]. Using this as a basis, it allows in priniciple to define the relationship between darts and cells. Probably [17] has a way of how to construct a cell from darts using involutions. I do not know if this is relevant for your study. But, the reverse certainly is: How to construct a dart from the cells of a cell complex! In my understanding, a dart is a cell tuple V->F->E->P, where V is a volume, F a face, E an edge and P a point, and where x->y means 'x is bounded by y' with respect to the boundary relation given by the complex. If this is the case, then please write it down in the manuscript. If this is not the case, I want to see an alternative rigourous definition. In any case, the relationship between darts and cells must be completely clarified. I do hope to have made this clear.

- "dart belongs to precisely one cell". As "belongs to" is an imprecise predicate, this statement does not make sense (cf. above). But what you want, is a map

b: Darts -> Cells

which takes a dart to some uniquely defined cell. If my definition of dart as a cell tuple is correct, then you can write e.g. "b maps dart (V->F->E->P) to cell V". In this way, the map b could be a definition for your usage of "dart belongs to cell". If you need this for cells of other dimensions, then

how about:

b_i: Darts -> Cells, (C_n->...->C_0) maps to C_i

where C_k are cells for k=0,...,n. This gives you a map for each dimension i.

- "dart with adjacent cell". This is another undefined statement. Here, I am at a loss and am really puzzled about the meaning of this statement.

- "\beta_i". What is a rigourous definition of \beta_i? My guess is that it is an involution operating on the set of darts, if i = 2 or 3. For i=1, I guess, it is any permutation of darts. My guess comes from the Wikipedia page.

- "sewing operation". Please also give a rigourous definition of that. In particular, the word "linked" is ambiguous, to the least.

My second concern is that of complexity. You argue that it cannot be stated in terms of cells, only of darts. I do not agree. The reason is as follows (assuming my cell tuple definition of dart): If a volume has at most f faces and at most e edges, then the worst-case number of darts is

O(V x f x e).

That is a statement about cells, and now I have some idea about the complexity of your algorithm in terms of the more familiar part of topology.

Minor concern: There is a reference [?}, which should be citing Hatcher's book, I think.

Please forgive my that I am the cause for asking you to make yet another major revision, especially of the part containing the definitions. But I really am looking forward to seeing this article published soon.

Author Response

Thanks for your reply and i have supplied the rebuttal, please see the attachment.
